# How to Discover Traditional Varieties and Shape in a National Germplasm Collection: The Case of Finnish Seed Born Apples (*Malus × domestica* Borkh.)

**Maarit Heinonen \*** and **Lidija Bitz**

Natural Resources Institute Finland (Luke), Production Systems, Plant Genetics, Myllytie 1, 31 600 Jokioinen, Finland; lidija.bitz@luke.fi

\* Correspondence: maarit.heinonen@luke.fi; Tel.: +358-29-5326117

**Abstract:** Cultivated apple (*Malus × domestica* Borkh.) is a major crop of economic importance, both globally and regionally. It is currently, and was also in the past, the main commercial fruit in the northern European countries. In Finland, apple trees are grown on the frontier of their northern growing limits. Because of these limits, growing an apple tree from a seed was discovered in practice to be the most appropriate method to get trees that bear fruit for people in the north. This created a unique culturally and genetically rich native germplasm to meet the various needs of apple growers and consumers from the late 1800s to the mid-1900s. The preservation, study and use of this genetic heritage falls within the mandate of the Finnish National Genetic Resources Program. The first national apple clonal collection for germplasm preservation was reorganized from the collections of apple breeders. The need to evaluate the accessions, both in this collection and possible missing ones, to meet the program strategy lead us to evaluate the Finnish apple heritage that is still available in situ in gardens. In this article we use multiple-approach methodologies and datasets to gain well-described, proof-rich samples for the trueness-to-type analysis of old heirloom apple varieties. The approach includes a combination of socio-historic, pomological and genotyping methods and datasets that are all valued as equally important. The main finding was that in addition to the pomological, molecular and genetic evaluation of ex situ apple collections, an extensive historical data and socio-economic conditions research are essential to perform good characterization of accessions. After implementing the results in re-creating the Finnish national apple germplasm collection, the number of Finnish local varieties was more than doubled from 38 accessions to 97.

**Keywords:** cultivated apple; genetic resources; in situ inventory; cultural knowledge; phenotyping; genotyping; sustainable use

## 1. Introduction

Cultivated apple trees (*Malus × domestica* Borkh.) are adaptable to cold-temperate areas, and are commercially grown even within Siberia and northern China, where winter temperatures can reach –40 °C [1]. In Europe, the main apple producers are Poland, Italy and France [2]. Apple cultivation in Finland represents the final frontier of apple's northern growing limits. The main present-day commercial production of apples in Finland is limited to the southwest of the country, which has the longest growing season and mildest winter climate. Similarly, in the past, the first apple trees were documented to have been grown in particular gardens of southern Finland in the 16th and 17th centuries [3–5]. In 1709, a winter frost destroyed almost all apple trees in Fennoscandia. Afterwards, Finns were encouraged to continue to grow apple trees. According to the statistics from the 1820s,

at this time there were nearly 4000 fruit gardens in Finland, two thirds of them in the southwest of the country [6]. One century later, in the 1920s, the estimated number of apple trees increased dramatically (1.68 million) and this number almost doubled only 10 years later (3 million trees at the end of the 1930s) [6]. Nowadays, in Finland, the commercial apple cultivation comprises about 670 hectares [7].

The first documented cultivated apple saplings in the 16th century were imported from Baltic countries (the very first ones were from Tallinn), and later from other neighboring countries (Russia and the south of Sweden) and countries south of Finland (Germany, Denmark and France). However, winter frosts, common and regular in Finland, regularly destroyed the first orchards [3,6,8,9]. In order to maintain apple production in freezing conditions, despite continuous destruction, a new solution was needed, and this solution relied on spontaneous selection of trees where only one characteristic was crucial: to be able to survive harsh conditions. Thus, growing an apple tree from a seed was discovered rather soon—already in 18th century—to be in practice the most appropriate method to get trees that can bear fruit for people in the north. Moreover, this practice was actively recommended by local agronomical experts of the time [8] up until the mid-1900s. Once Finns realized that it is possible to grow apples in Finland, the number of apple trees rapidly increased, as shown above.

Finns were forced to rely on different methods in order to be able to possess this valuable nutrition and vitamin source in their own gardens. For example, from oral heritage [10,11] and some historical written documents [3,6,8,12] we learnt that it was not unusual for people to sow the seeds from any apple fruit they could find, and by chance some of the seedlings would survive for many seasons and, in addition, give descent and usable fruits. This means of acquisition was also common because of the lack of local nurseries in Finland. Only wealthy farm estates (mainly manor estates) had contacts and could afford to purchase them from abroad [12]. Maintenance of seed-born apples that are able to be propagated season after season has created a unique, specifically locally adapted native apple germplasm. Diverse, culturally and genetically unique apples were selected for various needs (processing; fresh consumption also in wintertime, self-sufficiency in apples) that apple growers used from the late 1800s to the mid-1900s [3,6,8,12].

The Finnish breeding program for cultivated apple was established in 1958 on the basis of extensive pomological observations (in total, 315 apple varieties were studied from 1935 to 1958) [13] at the Agricultural Research Center (current Natural Resources Institute Finland) situated in Piikkiö in the southwest of Finland (60° 25′ N, 22° 31′E). The main breeding goal was to combine the winter hardiness and early maturing characteristics from Finnish seed born local varieties to the fresh fruit quality from foreign delicacy cultivars [13]. It resulted in 17 new apple cultivars of which 12 have the Finnish local apple variety as a parent. Later, in the mid-1990s, the breeding program was targeted for apple scab resistant cultivars and provided three new cultivars, one of which contains the germplasm of local varieties [14].

Collection, maintenance, evaluation and support regarding sustainable use of environmentally, historically and socially shaped and pruned plant genetic resources falls under the auspice of the individual countries and often under the mandate of program for plant genetic resources, if such a program exists. In the case of apple genetic resources, valuable genotypes are preserved, almost exclusively, in ex situ national collections. In general, and originally, fruit and berry gene banks were established as working collections for improvement and breeding program, to provide materials for fundamental plant research, to archive historical genotypes and for educational activities [15]. This was also the case in Finland. In 2001, the Finnish apple germplasm collection was established for genetic conservation purposes at the horticultural research station of Agricultural Research Centre Finland [4]. It comprised the collection established for the breeding program.

At this point we come to the first open question and one of our hypothesis: how were apples chosen to be placed into the national collection and how can we find them/how can the mother trees be found? Marking an apple as an heirloom cultivar, or another important cultivar for the inclusion into the apple gene bank (collection), is seldom based on genetic analysis [16]. This was especially not the case in the past. Genetic evaluation has been very extensively used during the last few decades and

many apple collections worldwide has been screened, e.g., in the United States of America [17] and China [18] as well as huge collections representing European apple germplasms [19–22]. However, these genetic identifications and evaluations were mainly performed on already formed and established collections in order to discard duplicates and describe genetic diversity. The second open question and hypothesis is: how is the trueness-to-type to be ensured in every particular case of inclusion of a certain apple variety and accession into the collection? Although genetic markers have proved to be useful for finding out duplicates and removing them from the collection, trueness-to-type is not possible without well-described references. Finding references, particularly in the case of historical apples is hard or even impossible. To ensure identity, morphological and pomological evaluation is usually done. Trueness-to-type identification based on only characteristic descriptions requires extensive experience and availability of large number of reference fruit samples and there is still a high risk of errors because of variation in fruits within the same variety regarding the age of the tree (a young tree may produce larger and more cover-colored fruits), diverse growing conditions, harvesting times and storage facilities.

Over 7000 apple cultivars have been described in the literature worldwide [1] and during the last few decades, tremendous effort has been made to genotype many regional, national and local apple ex situ collections. In genotyping studies, microsatellites were most often the markers of choice due to their high polymorphism, reproducibility, codominance and relative ease of analyses [23]. However, inter-comparative analyses among apples genotyped by different institutions, also meaning different equipment and experts, are not straightforward to align. Those comparative analyses would require the use of a large set of the same cultivars and the same microsatellite markers that would enable standardization of data. Researchers have tried to overcome this challenge in the grapevine SSR genotyping community by establishing a common *Vitis* International Variety Catalogue (VIVC) database containing fingerprints of 4192 cultivar genotypes by nine common microsatellites [24]. VIVC database contains 23,000 cultivars, breeding lines and *Vitis* species with various data beyond genotyping. It was created at the beginning of the 1980s as a database of grapevine genetic resources.

Following numerous examples from the region (Sweden, Latvia, Lithuania) [25–27] Finland also started to undertake wider genetic evaluations of apple collections [5]. After the first studies about Finnish apple collection were published, genotyping of new samples continued every year according to the need of various research projects and the Finnish National Genetic Resources Program.

In this ongoing work, a small group with various expertise in social sciences, pomology and genetics started to clarify a process and define an algorithm to reach the most informative sample in order to finally discover a trueness-to-type targeted apple variety for inclusion in the national apple germplasm collection. Recently, in rare research reports [21,22,28] attention has been paid to the quality of the samples. This does not mean the leaf quality or related technical issues, but properness of the sample for the trueness-to-type analysis. These studies pointed out that the majority of the samples do not constitute definite enough proof of the trueness-to-type of accessions.

In this article, different methodologies and datasets were combined into a unique iteration process for discovery of reference apple trees both in situ in-gardens and from ex situ collections in Finland that will become donors of samples for trueness-to-type analyses and finally confirm or reject final inclusion of accession into the national germplasm collection.

The analyses included: (1) evaluation of the existing ex situ apple collection with morphological and microsatellite genetic measures; (2) production of a list of potential candidates of local seed born apple names based on old pomologies and other documents; (3) evaluation of local socio-historical contexts and morphological characteristics of the targeted apple varieties; (4) a public call and in situ in-garden inventory for concrete apples and local history knowledge; and (5) microsatellite genotyping of selected samples where all factors are valued as being equally important in gaining the ultimate result, i.e., the true-to-type varieties and accessions. Our approach had general parts and more specialized parts, the latter are dependent on local climatic, environmental and geo-historical conditions. We considered that, with the modification of specialized parts, this process might

have global applications. The missing links and gaps in the process were identified and solutions were proposed.

## 2. Materials and Methods

Local varieties, born from seed in Finland and spread to local or even broader cultivation, were chosen for the study according to the guidelines set by the Finnish National Genetic Resources Program [29] that aims to conserve, study and promote the sustainable use of agri- and horticultural plants, especially those of Finnish origin.

### 2.1. Finnish National Apple Ex Situ Collection

In the years 2002–2005, the Finnish apple collection consisted of 242 accessions: of which 54 are of Finnish origin, 95 Russian, 32 North-American, 22 Scandinavian, 16 Baltic, seven Central European and two Japanese [4]. The group of 54 apple varieties (or accessions) of Finnish origin consisted of 16 Finnish bred cultivars; 38 accessions were categorized as "unknown seedling from Finland", i.e., Finnish seed-born local varieties. From the Finnish apple collection, young leaves were sampled for the purpose of the microsatellite genetic analyses while fruits were used for morphological identification of varieties.

### 2.2. Defining Gaps and Verification of Identity of Accessions Existing in the Collection

The first evaluation of possible Finnish apple collection gaps was based on the descriptions of foreign and local apple varieties available in the Finnish pomology from the 1940s written by Olavi Meurman [3]. Other available pomologies [6,8,12,30] and old written documents from different sources (in Finnish and Swedish) dating from the late 1800s to the mid-1900s were also used. Based on historic origin information, morphology descriptions, evaluation of the use value, and agricultural value, we were able to start building a list of local seed-born apple varieties with known names potentially missing in the collection, representing historic apple heritage. Moreover, we were able to find out data about apples that should be additionally verified to confirm the identity of the apples already existing in the collection. These were added to the list.

### 2.3. Evaluation of the Apples

Based on a literature overview we assembled a set of characteristics that need to be obtained for each missing apple both from the old literature and during the following in situ in-garden inventory: (a) location of the tree (municipality, village, street and location in the garden); (b) remembered name of the tree (including synonym names); (c) estimated age of the tree; (d) acquisition (seedling, crafted, sapling); (e) acquisition site (e.g., market place, garden, nursery); (f) morphological characteristic of the fruit (color and shape, analyzed according to UPOV descriptors for the cultivated apple [31]); (g) phenological characteristics (maturing time, analyzed according to UPOV descriptors for the cultivated apple [31]). In addition to those characteristics we also collected old and new photos and illustrations the apple trees and fruits, and various socio-historical data, i.e., names and memorized data of the persons linked to the tree (especially persons who sewed the seed or planted the tree in the past, or persons who were the first to recognize the value of the tree), of their original sites and of the distribution route from the original site to other gardens.

### 2.4. In Situ In-Garden Inventory

Based on the list of potential Finnish seed-born apple varieties, both those that were missing from the collection and those that already existed in the collection, a country-wide public call (through various media) was released to announce the search in situ in-garden. Collated data was followed by the descriptors for national in situ landrace inventories [32].

Based on call outcomes, we performed a series of in situ in-garden inventories in order to find the living trees representing targeted apples and sample them for further trueness-to-type analyses. The inventory was targeted to mother trees, trees that were over 50 years old, or young trees with well documented grafting history to avoid receiving samples from trees purchased from present-day nurseries.

The updated call is repeated yearly to collect samples from as many targeted apples as possible, including leaves and fruits, and traditional knowledge possessed by the owners. The results are evaluated according to the assembled set of characteristics. The process of in situ in-garden inventory is presented as the part of the landrace in situ conservation strategy for Finland [11] and has followed in the general guidelines for creating national inventories of on-farm genetic diversity recommended by European Cooperative Programme for Plant Genetic Resources (ECPGR) [33].

We also sampled apples of the named varieties from nurseries in Finland and some from the national apple germplasm collections of neighboring countries (Estonia, Sweden) to be used as reference samples.

*2.5. Microsatellite Fingerprinting*

Accessions from the Finnish collection and the most promising candidates for missing apples and verification of apples were selected for microsatellite marker analysis. Our aim was to obtain at least two in situ samples of the named varieties to compare with each other and/or with accessions in the apple collection and other apple variety reference data.

A set of eight microsatellite markers [Ch02c06, COL, Ch04e05, Ch01h02, Ch02c11 and Ch02c09 [34] CH02d08, Ch01g12] were used for initial analyses of Finnish apples considered to be included into newly established national collection. During the process, the fingerprinting analysis has been extended with eight additional markers [Ch01f02, Ch01f03b, Ch01h01, Ch01f02, Ch04c07, GD12, GD147, Hi02c07]. The markers were selected from previously published studies [35,36]. DARwin software was used for calculation of distance measure based on a dissimilarity simple matching coefficient from allelic data and construct tree by weighted neighbor-joining method. Alignment of all fingerprints and inter-comparative analysis among them resulted in identification of duplicates, which, in our case, was evidence for confirmation of trueness to type.

## 3. Results

*3.1. Names, Origin, Acquisition, Historical and Characterization Data of Finnish Local Varieties*

The newly-established Finnish apple collection contained 242 accessions of diverse origin where the majority of them were of Russian (95) and then Finnish origin (54), of which 38 were categorized as Finnish seed-born local varieties. During the evaluation and diverse assessment of the collection, a group of researchers realized that more evidence would be needed to ensure the identity of apple varieties (trueness-to-type) as well as to identify gaps that have been determined in the collection.

The very first steps were hard to identify, i.e., how to find argumentation that we indeed have trueness-to-type accessions in the collection while we were lacking the proper methodology. By the old pomological literature and historical documents overview we were able to do this. Therefore, an extensive literature overview was started to find and gather data about old apples from Finland and memorized data relating to those apples, potentially representing gaps and clues for obtaining samples to verify identities.

By overviewing mainly Finnish pomologies, we were able to set up a list of 80 Finnish seed born apple varieties. Meurman's pomology dating from the 1940s was especially valuable because he concentrated on describing almost only local apple varieties: 40 local varieties of Finnish origin, 87 local varieties of foreign origin, and only nine cultivars of foreign origin. In parallel, all other kind of available data (putative origin, site of growing/cultivation, descriptions) about listed apples were

collected from the literature and that will also serve later during the public call for apple and following inventory in situ in gardens.

In order to be able to perform in situ in-garden inventory to find targeted apples, a tool for locating them was needed. For this purpose, we developed a public call for apples including as much information as we could get about desired local apple varieties. Since many Finnish local apple varieties have become decidedly rare, we needed to develop a specific method also to reach those who know the growing history and habitat of the targeted variety. The national calls for named apple varieties were released via numerous local and national print, audiovisual and social media as well as public events with short announcements, articles and posters.

Tree owners were asked to contact the research group with as much detailed evidence as possible as follows: (a) remembered name and estimated age of the tree; (b) acquisition method (seedling, crafted, sapling),time (e.g., decade) and site (e.g., from a manor garden or nursery nearby); (c) location (garden address, tree's location in garden); (d) morphological (mainly fruit characteristics: color, shape) and maturing time details (summer, autumn or winter variety). They were encouraged to provide also information of fruit (use as fresh, in cooking, storage durability, taste), memories linked to the tree (e.g., person how sowed the seed or planted the sapling), and to share new and old photos and other documents (e.g., garden schemes, local newspaper articles, exchange of letters). Part of the results from this intensive inventory is presented in Table 1. We selected 12 apple varieties/accessions representing some of the most typical challenges we faced during the process.

The received announcements were evaluated against all information received from various sources, e.g., pomology descriptions and other already gathered historic documents, e.g., manor estates and their owners' interest on apple cultivating. One very important reference data set used in the evaluation the sample's value was the extensive photo gallery of fruits of apple varieties, both modern and traditional ones, pictured in different growing conditions. Another absolutely necessary reference data set was the large variety of fresh fruit samples available in autumn from apple collections to be able to observe morphological details applying UPOV standards of received fruit samples.

During the next step in the multi-approach trueness-to-type process, i.e., the in situ in-garden inventory, an additional 20–30 apple names were added to the list of named varieties to search for. These were not described in the main pomology references (as they concentrated on evaluating the most proper ones for promoting apple cultivation of the time) but met the definition of a Finnish local variety, i.e., they were believably memorized as seed-born and spread to local cultivation (e.g., they were included in nursery catalogues).

The first inventory phase was carried out from 2011 to 2014 when 268 samples were selected from over 500 notifications for molecular analysis. It was necessary to prolong the inventory time for better coverage. The present-day (year 2018) sampling data of Finnish apple varieties (obsolete varieties, local varieties) and traditional varieties consists of 1000 samples in total.

The best-known local varieties which had spread widely in cultivation were easy to find in situ in-garden, and substantially documented historic origin data available. There were no contradictions within morphological and genotype identifiers among their samples. This was the case, e.g., with 'Huvitus' and 'Grenman'. The most prominent challenges during the gap filling process and finding of all targeted apples could be classified into several distinct groups, i.e., cases that were particularly hard to solve (Table 1.): (a) diploid-triploid pairs; (b) renamed foreign varieties; (c) several candidates for trueness-to-type; (d) very rare local variety; and (e) trueness-to-type mother tree (i.e., the original seed-born tree in its original site). The types are presented here by providing cases from the inventory data.

**Table 1.** Names, origin, acquisition, historical and characterization data of 12 apple varieties classified into five cases in order to describe evaluation process for including samples into Finnish national apple collection.

| Part I | | | | | |
|---|---|---|---|---|---|
| **Type** [1] | **Type 1: Diploid-Triploid Pairs** | | | | **Type 3: Several Candidates for Trueness-to-Type** | |
| **Name** [2] | **Kavlås** | **Turso** | **Grenman** | **Eppulainen** | **Jalmarin talvi** | **Jalmari Karimaa** |
| **Synonym name** | Kavlås Kaflås, Kafelås, Kavelås, Kavelås' in omena | Turso, Tursa | Grenman, Grenmanin omena, Grenman omena | Eppulainen, Eppu, Einolainen | Jalmari, Jalmarin talvi, Jalmarin omena | Jalmari |
| **Age** [3] | Early 1800s | Early 1900s | Mid 1800s | Seed sown in 1925 | Late 1800s | Late 1800s |
| Location | | | | | | |
| *Original site* [4] | Swedish origin | Unknown. Possibly in Kangasala, Finland | Mikkeli, Finland | Kangasala, Finland | Asikkala, Finland | Asikkala, Finland |
| *In situ in-garden* [5] | Some announcements from home-gardens, commercial orchards and nurseries. 4 samples in SSR analysis of which 2 are from nurseries | Several announcements from home-gardens in the area, a few from commercial orchards. 13 samples in SSR analysis | A large number of announcements from home-gardens around Finland, a few from commercial orchards. 8 samples in SSR analysis, of which 1 was from the original site | A few announcements from home-gardens. 3 samples in SSR analysis, of which 1 was from the original site | A few announcements close to the origina site, a few from other parts of Finland. 7 samples in SSR analysis, of which 1 was from a nursery | 1 sample in SSR analysis |
| Acquisition way and site [6] | | | | | | |
| *Original site* | Probably seed born | Probably seed born | Most probably Seed-born | Seed-born. Grenman's seedling | Seed born | Seed born |
| *In situ in-garden* | Saplings available in nurseries, especially in central Finland from the early 1900s | A preacher sold saplings/grafts door-to-door in the early 1990s. Saplings in nurseries in the late 1900s | Saplings available in many parts of Finland, especially in central and northern Finland from the early 1900s | Saplings in nurseries in the late 1900s | Locals grafted from the mother tree, since 1940s on saplings available first in local nurseries and later in some other nurseries | Jalmari named person from the original site in Asikkala moved to western Finland and brought grafts/saplings in the mid-1900s |
| Local history [7] | | | | | | |
| *Original site* | Rich local history in Sweden | Rather limited local history | Rich local history | Some local history | Rather limited local history | Very limited local history |
| *In situ in-garden* | Rich local history especially in southern and central Finland | Some local history, especially in central Finland | Rich local history especially in southern and central Finland | Some local history especially in central Finland | Rather limited local history in southern Finland | Very limited local history |
| Characterization [8] | | | | | | |
| *Plant habit* | Ramified: 2 | Ramified: 3 | Ramified: 1 | Ramified: 2 | Ramified: 2 | Ramified: 2 |
| *Time* | Flowering: 5; Eating maturity: 6 | Flowering: 5; Eating maturity: 5 | Flowering: 3; Eating maturity: 6 | Flowering: 5; Eating maturity: 6 | Flowering: 7; Eating maturity: 9 | Flowering: no data; Eating maturity: 7 |
| *Fruit* | Shape: 2; Over color coverage: 1; Over color: 4; Over color type 5 | Shape: 6; Over color coverage: 5; Over color: 3; Over color type 7 | Shape: 7; Over color coverage: 5; Over color: 5; Over color type 3 | Shape: 7; Over color coverage: 5; Over color: 3; Over color type 3 | Shape: combination of 2 and 7; Over color coverage: 5; Over color: 1; Over color type 7 | Shape: combination of 4 and 6; Over color coverage: 5; Over color: 1; Over color type 7 |

**Table 1.** *Cont.*

| Part I | | | | | |
|---|---|---|---|---|---|
| **Type** [1] | **Type 1: Diploid-Triploid Pairs** | | | | **Type 3: Several Candidates for Trueness-to-Type** |
| *Photo* [9] |  |  |  |  |   |

| Part II | | | | | |
|---|---|---|---|---|---|
| **Type** [1] | **Type 2: Renamed Foreign Varieties** | | | **Type 4: Very Rare Local Variety** | **Type 5: True-to-Type Mother Tree** |
| **Name** [2] | **Långsjön päärynäomena** | **Korobovka** | **Louhisaaren ananas** | **Tallinn pirnõun** | **Mannerheimin omena** | **Huvitus** |
| **Synonym Name** | Långsjön päärynäomena, Lånsjöns päronäpple, Sagulinin päärynäomena, Sagulins päronapple, Esterin päärynäomena, Esters päronäpple | Korobovka | Louhisaaren omena, Willnäs ananas | Tallinnan päärynäomena, Räävelin päärynäomena, Rääveliläinen päärynäomena, Revals päronäpple, Revalskt päronäpple, Långs Revals päronäpple, Revaler Birn-apfel | Mannerheim omena, Mannerheim | Huvitus, Korpelan omena |
| **Age** [3] | Late 1800s in Somero | Not known. Arrived to Finland from Russia in 1930s | Early 1800s in Louhisaari | Late 1700s | Late 1800s | Seed sown in 1895. Tree alive. |
| **Location** [4] | | | | | | |
| *Original site* [d] | Named in Somero, Finland | Possibly Turkish origin | Named in Louhisaari, Finland. | Estonian origin, possibly in Tallinn area | Asikkala, Finland | Yläne, Finland |
| *In situ in-garden* | A few announcements from home-gardens. 5 samples in SSR analysis, of which 1 was from the original site. | No announcements from home-gardens. 1 sample in SRR analysis as a reference sample from the Estonian apple germplasm collection. | A few announcements from home-gardens. 7 samples in SSR analysis, of which 1 was from a nursery | Several announcements from home-gardens in the south of Finland. 4 samples in SRR analysis, of which 1 was a reference sample from the Estonian apple germplasm collection. | Very few announcements in home-gardens. 9 samples in SSR analysis | A large number of announcements from home-gardens around Finland, a few from commercial orchards. 6 samples in SSR analysis, of which were from the original site (including 1 from the rootstock) |
| **Acquisition way and site** [5] | | | | | | |
| *Original site* | Grafts or saplings brought from Russia (St. Petersburg) | Probably seed-born | Grafts or saplings brought from Germany | Most probably seed born | Seed born | Fruit seeds originate from an old derelict orchard nearby. |

**Table 1.** *Cont.*

| Part II | | | | | |
|---|---|---|---|---|---|
| **Type** [1] | **Type 2: Renamed Foreign Varieties** | | | | **Type 4: Very Rare Local Variety** | **Type 5: True-to-Type Mother Tree** |
| **Name** [2] | Långsjön päärynäomena | Korobovka | Louhisaaren ananas | Tallinn pirnõun | Mannerheimin omena | Huvitus |
| ***In situ in-garden*** | Saplings available mainly in local nurseries, to some extent in southern Finland from the mid-1900s | Saplings available in some nurseries, especially in eastern Finland since the 1930s | Locals crafted from the old tree, since the late 1900s available in some nurseries | From early 1900s, saplings available in nurseries | Very few saplings in the area, most probably grafted from the garden owner who recognized the value of the mother tree | Locals crafted from the mother tree, since the 1910s/1920s, saplings available in local nurseries, since 1940s, saplings available in nurseries in Finland |
| Local history [6] | | | | | | |
| ***Original site*** | Some local history | Not known | Some local history | Rich local history in Estonia | Rather limited local history | Very rich local history |
| ***In situ in-garden*** | Some local history especially in southern Finland | Very limited local history | Some local history especially in southern Finland | Rich local history especially in southern Finland | Very limited local history | Very rich local history in many parts of Finland |
| Characterization [7] | | | | | | |
| ***Plant habit*** | Ramified: 2 | Ramified: 2 | Ramified: 2 | Ramified: 1 | Ramified: 1 | Ramified: 2 |
| ***Time*** | Flowering: 5; Eating maturity: 4 | Flowering: 5; Eating maturity: 4 | Flowering: 5; Eating maturity: 5 | Flowering: 5; Eating maturity: 5 | Flowering: 3; Eating maturity: 4 | Flowering: 2; Eating maturity: 2 |
| ***Fruit*** | Shape: 7; Over color coverage: 3; Over color: 5; Over color type 5 | Shape: 7; Over color coverage: 3; Over color: 1; Over color type 5 | Shape: 6; Over color coverage: 5; Over color: 2; Over color type 2 | Shape: 2; Over color coverage: 5; Over color: 5; Over color type 1 | Shape: 6; Over color coverage: 1; Over color: 3; Over color type 7 | Shape: 5; Over color coverage: 5; Over color: 4; Over color type 7 |
| ***Photo*** [8] |  | N.A. [9] |  |  |  |  |

[1] Type is related to the group of the cases we described to illustrate process of shaping national apples collections; [2] Name of apples used in the database of the Finnish apple germplasm collection. Korobovka and Tallinn pirnõun are synonymous names in the database. Jalmari Karimaa is a sample name of the in situ in-garden inventory and not in the database; [3] Estimated age of the mother tree or very old tree used as a source for grafting; [4] Original location refers to the mother tree or very old tree used as a source for grafting. It may not be found in the site anymore; [5] In situ in-garden refers to locations/areas of the inventory data received (announcements, samples); [6] Acquisition ways are seedling (seed sown), grafted from an older tree, sapling from e.g., nursery. Acquisition sites of seed (i.e., fruits whose seeds are sown), grafts, and sapling (e.g., marketplace, garden, nursery). Original site refers to the mother tree or very old tree. In situ in-garden refers to the samples of in situ in-garden inventory; [7] Local history and knowledge refers to persons linked to the tree (e.g., persons who sowed the seed, did the crafting, planted the sapling; persons who recognized the value of the tree), sites (e.g., apple cultivation history in the area, i.e., region, municipal, village, garden); written documents (e.g., pomology description, garden schemes, local newspaper articles, exchange of letters); use value (use as fresh, in cooking; storage durability, taste, etc.). Here, we presented the rough volume of local knowledge available and documented; [8] Characteristics observed from fruit samples of in situ in-garden and the apple collections. Plant habit observed from accessions in the national apple germplasm collection. Used UPOV standards for *Malus domestica*; [9] Visual data contains present-day and old photos (of fruits, tree, garden, persons linked to the tree, etc.). Here we provide one present-day photo to illustrate morphological characteristics of the fruit; [9] Not available.

### 3.2. Microsatellite Fingerprinting

At the beginning, the goal was to produce a microsatellite fingerprint of each accession and start building evidence about (a) accessions with unrepeated profile, (b) different accessions having identical profile (duplicates) and accessions with the same name but divergent genetic profiles. The database grew with each year of surveys, i.e., the number of samples and markers increased and we were also able to define trueness-to-type apples (those represented by a higher number of samples obtained from different regions with the same name, and identical genetic profiles). These are nowadays used as reference cultivars in genotyping along a set of unique fingerprints that we assume to be a pool of all unrepeated genetic profiles, with repeated genetic profiles being represented only once and true-to-type profiles being represented only once.

### 3.3. Combining Morphological, Fingerprinting and Historic Origin Identifiers for Obtaining Trueness-To-Type Accessions

Every year we analyzed the received results of microsatellite analysis in the context of historic data and characteristic identifiers and compared them to all genotype identifiers of earlier years. In research data, we identified several samples of the named varieties which required us to iterate the analysis and collate further data for the trueness-to-type analysis. The types are presented here by providing cases from the inventory data.

#### 3.3.1. Diploid-Triploid Pairs

During the analyses of microsatellite fingerprints, we noticed that some accessions, e.g., Turso and Eppulainen, sometimes have three microsatellite alleles in the frame of one marker, which might be an indication that those accessions are triploids (Table 2.) This was already recognized when we analyzed a triploid variety, 'Rambo' [37]. Including polyploidy samples to the dendrogram caused misinterpretations in defining genotype identifiers to the trueness-to-type varieties. In two cases of the microsatellite data of 1000 samples, two morphologically distinctive varieties (Table 1) had the same genotype identifier, i.e., were shown to be duplicates.

The first diploid-triploid pair was 'Turso' and 'Kavlås' which both produce very large fruits. According to historic documents, a preacher sold saplings to finance his evangelist journeys in Finland in the early 1900s. These saplings were named 'Turso', but its original site has remained unclear. It is presumed to originate from in Kangasala municipality (61° 27'N, 24° 04'E) in a homestead which has a rather similar name, 'Turso', but no oral or written knowledge was found to support that. 'Kavlås' is a Swedish traditional autumn variety dating back to the 1830s. In the 1940s, it was recommended to cultivate this variety, especially in the central parts of Finland, because of its exceptional winter-hardiness properties.

The first results of microsatellite analysis containing samples named 'Turso' was interpreted as a synonym name of 'Kavlås' because they shared the same genotype results. However, the result was in contradiction with morphological identifiers and historic data. A targeted call for old apple trees named 'Turso' in the area of its supposed original site (Kangasala municipal) was released to get additional local knowledge and samples of apple trees to compare. The call resulted in several responses from tree owners. Five of them were selected for microsatellite analysis to be compared to samples obtained from nurseries and all other genotype data. They were all confirmed to have an identical genotype identifier, which was the same as 'Kavlås'. However, the new 'Turso' named samples also showed contradictions in morphological identifiers compared to 'Kavlås'. The result of ploidy status analysis verified them as separate varieties: 'Kavlås' has a diploid while 'Turso' has a triploid genome size (Table 2). While 'Turso' was verified as distinctive from 'Kavlås' and was not identical to any other named variety, it is most probably Finnish origin. Historic data shows no evidence of 'Turso' being a 'Kavlås' seedling but in principle that might not be impossible because winter hardy 'Kavlås' is known to grow in Finland from late 1800s and as well in the central parts of Finland where 'Turso' is suspected to originate from.

**Table 2.** Microsatellite fingerprints for diploid-triploid pairs 'Kavlås' and 'Turso'; 'Grenman' and 'Eppulainen'; shown only for markers that had more than two alleles.

| True to Type Name | CH02c06 | CH02c06 | CH02c06 | CH02c09 | CH02c09 | CH02c09 | CH02c11 | CH02c11 | CH02c11 | CH02d08 | CH02d08 | CH02d08 | CH04e05 | CH04e05 | CH04e05 | CH01h02 | CH01h02 | CH01h02 | COL | COL | COL |
|---|---|---|---|---|---|---|---|---|---|---|---|---|---|---|---|---|---|---|---|---|---|
| | | | | | | | | | Kavlås and Turso | | | | | | | | | | | | |
| **KAVLÅS** (diploid *) | 219 | 232 | - | 242 | 244 | - | 217 | 219 | - | 214 | 227 | - | 207 | 210 | - | 242 | 250 | - | - | 233 | 242 |
| **TURSO** (triploid *) | 219 | 232 | - | 242 | 244 | **250** | 217 | 219 | **225** | 214 | 227 | **249** | 207 | 210 | **228** | 242 | 250 | - | **231** | 233 | 242 |
| | | | | | | | | | Grenman and Eppulainen | | | | | | | | | | | | |
| **GRENMAN** (diploid*) | 204 | 232 | - | 244 | - | 250 | 217 | 219 | - | 227 | 227 | - | 175 | 212 | - | 238 | 242 | - | 231 | 233 | - |
| **EPPULAINEN** (triploid *) | 204 | 232 | **245** | 244 | **246** | 250 | 217 | 219 | - | 227 | - | **248** | 175 | 212 | **228** | 238 | 242 | **250** | 231 | 233 | - |

* To validate the morphological and microsatellite data the ploidy status of set of apples was determined by flow cytometry (Plant Cytometry Services, Schijndel, The Netherlands).

The second diploid-triploid pair has been documented to have a family relationship. According to historic origin data, 'Eppulainen' is a seedling of 'Grenman' sown in 1925 in Kangasala municipality (61° 27′N, 24° 04′E). 'Grenman' originates from Mikkeli municipality (61° 67′ N, 27° 19′ E) in the 1850s and has been a popular variety since the early 1900s. The result of ploidy status revealed the difference: 'Grenman' is a diploid and 'Eppulainen' is a triploid variety (Table 2). Their distinctiveness was also verified at the genetic level.

### 3.3.2. Renamed Foreign Varieties

Already in the 1940s it was suspected that several varieties with Finnish names were actually renamed foreign varieties. Applying the multi-approached trueness-to-type approach had advantages also in identifying renamed varieties and revealing their geographical origin. In the late 19th century, manor estates in Finland imported young trees from neighboring countries, especially from Russia and Sweden. Those that showed some adaptability to local growing conditions were selected for grafting. Many manor estates established nurseries to sell their own produce or imported saplings. Typically, saplings were renamed in Finnish or Swedish since most manor owners' mother tongue was Swedish. Mainly renamed ones were translated names, for example the Baltic traditional variety 'Transparente blanche' was assigned the word-by-word translated names 'Valkea kuulas' and 'Valkoinen kirkasomena'. In these cases, the country or geographical origin can rather easily be understood from the translated name. Translating was a common habit in many countries because foreign names were felt to be difficult for consumers at that time. In some cases, original names of trees may had been forgotten but obviously renaming had also offered nurseries the possibility of branding by including their nursery name in the variety name.

In 1887, a Finnish pomologist recognized an apple tree that would be valuable for distribution in a manor orchard in Somero municipality (60° 38′ N, 22° 54′ E). It was ordered as a sapling or crafts from the famous Regel nursery in St Petersburg in Russia. He named it 'Sagulinin päärynäomena' referring to the manor owner's surname (Sagulin) and the aroma of the fruit (päärynäomena is pear apple in English). However, that name was not brought into use among nurseries, who instead used the name given by the manor owner who had a nursery and horticultural school: 'Långsjön päärynäomena' ('Långsjöns päronapple' in Swedish). Långsjö refers to the Swedish name of the manor estate. It has also been marketed with the name of 'Esterin päärynäomena' ('Esters päronäpple' in Swedish). Its small fruits made it a rather popular variety in home gardens because of their special aroma. Later, Finnish pomologists suspected it to be identical to the traditional Belorussian variety 'Korobovka' because of its similar fruit and other characteristics.

Microsatellites of samples named 'Långsjön päärynäomena' and 'Esterin päärynäomena' were compared to a sample named 'Koropovka' received from the Estonian germplasm apple collection at the Polli Horticultural Research Center. Analysis showed the very same genotype identifier for all those samples (Figure 1) and verified them as synonymous names. The variety with the Finnish accession name is accepted into the national germplasm apple collection as a valuable traditional foreign variety because of its long cultivation history in Finland. The fact that it is included in present-day nursery catalogues with a Finnish name shows its durable use value.

The second case of renamed foreign varieties had similarities with the first case. 'Louhisaaren ananas' was described by the first Finnish pomologist as originating from a tree brought to the manor in Louhisaari (60° 57′N, 21° 83′E) from Germany in the early 1800s. Similar to 'Långsjön päärynäomena' it has several synonymous names. Efforts made in the late 1900s to get it to nursery production were not successful until several decades later. Observations of the variety in apple collections at different sites casted doubts on whether it was identical to an Estonian traditional variety 'Tallinn Pirnõun' (several translated names are in use in Finland for this variety, which has had a long cultivation history in Finland; in 1929 it was one of the most cultivated varieties).

Microsatellites of samples named 'Louhisaaren ananas', samples of the translated names of 'Tallinn Pirnõun', were compared to a sample named 'Tallinn Pirnõun' received from the Estonian

germplasm apple collection at Polli Horticultural Research Center. Analysis showed that the genotype identifier of all the samples was identical (Figure 1), and hence they were verified as synonymous names. The result was not totally in contradiction with local historic data, i.e., a tree could have been shipped from Germany to Louhisaari manor because Germans have had a strong influence in Estonia in the past, and there is a high probability of plant material exchanges. Similar to 'Långsjön päärynä', it was accepted to the national apple clonal collection with the same criteria. Furthermore, it is in present-day nursery catalogues with both Finnish names. However, it remained unclear whether the trees for crafting were changed during the 1900s in Louhisaari manor, and that caused the confusion.

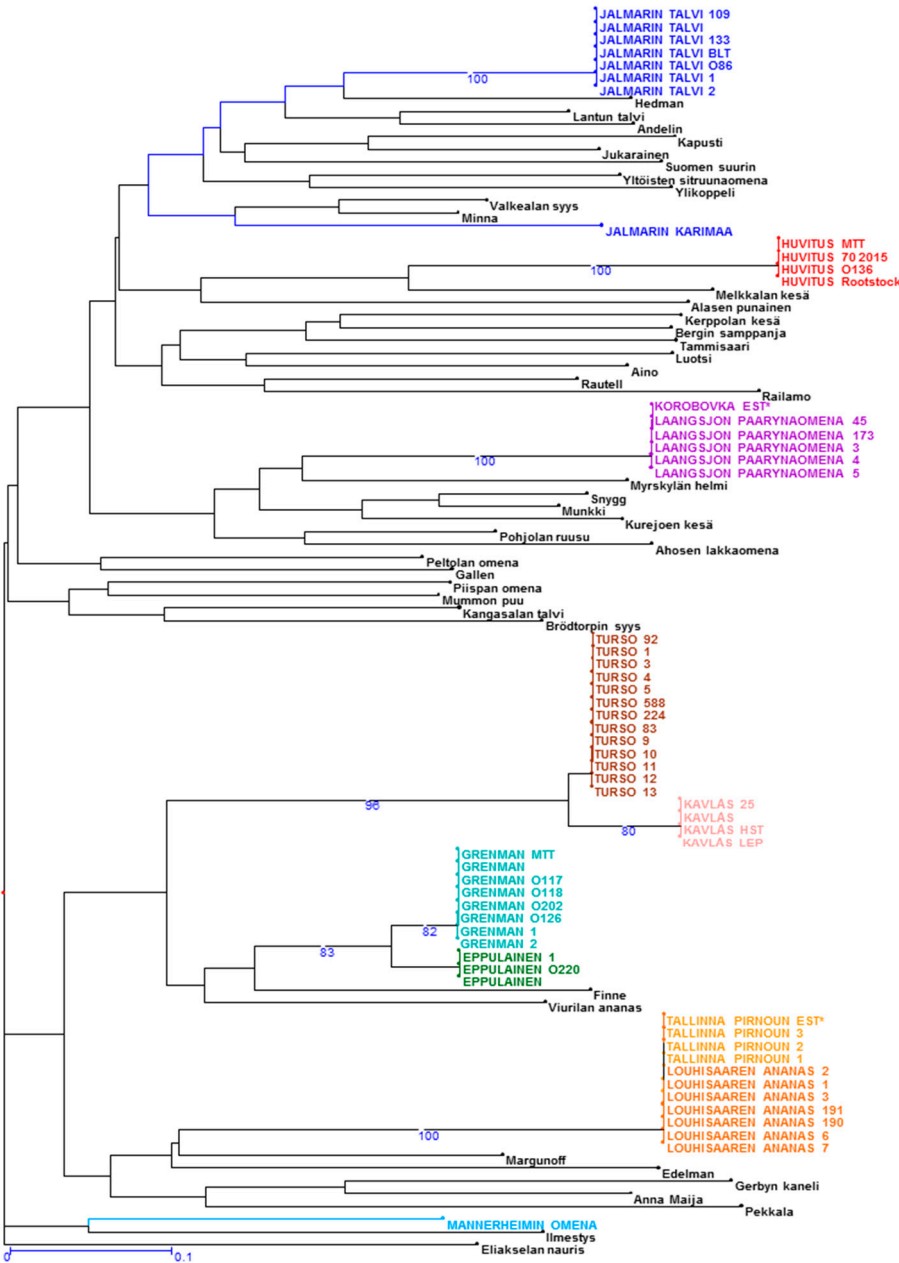

**Figure 1.** Dendrogram containing 50 Finnish true-to-type apple varieties with one wrongly named sample, Jalmari Karimaa. * Reference DNA for the cultivars Tallinn Pirnoun and Korobovka was obtained from the Estonian national germplasm apple collection at Polli Horticultural Research Centre.

### 3.3.3. Several Candidates for Trueness-to-Type

Microsatellite analysis results showed that we received numerous (every sixth sample) wrongly named candidates for trueness-to-type. One of these cases occurred during the trueness-to-type analysis of a local variety called 'Jalmarin talvi'. It was described and recommended for cultivation in the horticultural professional journal in 1941, just after the large winter damage of the winter 1939–1940, when there was an urgent need for new winter hardy varieties. The mother tree of 'Jalmarin talvi' in Asikkala municipality (61° 13′ N, 25° 69′ E) was one which had survived. According to knowledge from oral tradition, the seed was sown in the late 1800s. It has good storage durability even until spring, which was a very valued characteristic until the mid-1900s, as fresh apples could be supplied all year round. The variety name refers to the name of the farmer of the mother tree (Jalmari) and the maturing time (talvi is winter in English) shows that it belongs to the group of winter varieties.

Neither the genotype nor morphological identifiers were similar to the samples named 'Jalmarin talvi'. According to historic data, the mother tree was damaged some decades ago and died. The targeted call to the area close to its original site resulted in announcements from one garden with one old tree and one garden having five old trees called 'Jalmari'. The latter garden had obtained saplings about 70 years ago from the farm estate where the mother tree was. They all had an identical genotype identifier and had similarities in their morphological identifiers. Their genotype identifier was the same as the one already studied in the nursery sample but differed from the other sample (in Figure 1, sample name Jalmari Karimaa). The trueness-to-type ones and the wrongly named one sample with a unique genotype identifier were identified. One of the new samples of 'Jalmari' from the western part of Finland close to the sampling site of the Jalmari Karimaa was a trueness-to-type 'Jalmarin talvi'. The result revealed the probability of a grafting error in the past.

### 3.3.4. Verifying a Very Rare Local Variety

Two thirds of the identified potential Finnish local apple varieties (names of the varieties) were found in Finnish nursery catalogues dating from the 1800s to 1950. They have been actively in use and it is possible to locate old trees in several old gardens in different parts of Finland. Some varieties were only available for a short time and were only in sapling production in one or two nurseries. Hence, they are rare in the present-day gardens and it required very special efforts to locate them in gardens.

'Mannerheimin omena' was one of the local varieties we searched for. It was not described in the old Finnish pomologies. Instead, we found a very short news article in a horticultural professional magazine published in 1921. However, the four sentences provided core information for its historic origin: (a) location (lake island in Asikkala municipality), acquisition means (seed sown); (b) morphological characteristics (not-so-small fruit, yellowish color of skin); (c) persons linked to the tree (surname of the person who sowed the seed, name of the person who recognized its use value); (d) how it was named (hints of naming process). Furthermore, the name of the variety was found during the years 1930 and 1931 in one nursery catalogue.

Based on this information, the local heritage association was asked to use their local social networks in order to find further knowledge of the variety and old trees of that name. The targeted call resulted in one announcement from a former schoolmate of the two brothers in Asikkala (61° 28′N, 25° 62′E). The schoolmate was not familiar with the apple names but has had a fruit to eat from a tree named 'Mannerheim'. The tree owners' interviews (two brothers and one sister) provided strong proof that it is the local apple variety we searched for, although not the mother tree. We also received a color photo taken in the 1970s from the old lady in the manor estate where the person who recognized the value of the seed-born tree lived after retiring and planted one sapling of 'Mannerheimin omena' to the manor garden. She verified the morphological identifiers of fruits: yellowish rather large and rounded shape fruits. Unfortunately, the tree was cut down in the early 1980s because of a new public road.

The genotype identifier of the sample was unique compared to all genotype sample data, i.e., it was not any named known variety. As the Finnish genotype reference data can be regarded as extensive and oral traditions were not conflicting, we concluded that it is most probably the trueness-to-type one.

After announcing the findings in the national media to celebrate Finland's former president C. G. E Mannerheim's 100-year anniversary, some relatives of the core persons linked to the mother tree (one who sowed the seed and one who recognized the tree) contacted the research group providing further historic information. The oral tradition of the both families revealed details of how a seed-born tree was named after Finland's future president. Because the families had no apple trees of that name, once again a targeted call for old trees of the name 'Mannerheimin omena' was released (this time in the local newspaper) to get more samples for analysis. It resulted in several announcements but very weak historic origin proof. It was not a surprise that the samples' genetic identifiers were the same as already known varieties and none of them had the same identifier as the Mannerheimin omena sample.

The multi-approach trueness-to-type process for searching for proof to having the trueness-to-type 'Mannerheimin omena' fascinated many apple enthusiasts and the variety has retaken to nursery production. The process upgraded the cultural and social value to the local variety. In addition to belonging to the national germplasm apple collection it will be safely backed up in many home gardens and in the historic garden of Finland's former president Mannerheim's childhood manor estate.

3.3.5. Trueness-to-Type Mother Tree

Based on historic data of old pomologists and oral tradition of tree owners, we identified 10 mother trees. One of those was the most famous, cultivated and used one in apple breeding. In the 1940s, 'Huvitus' was described as a new potential local variety with exceptional combinations of winter hardiness, early maturing and good taste for fresh use. Later it was taken to the cultivated apple breeding program and 'Huvitus' is a parent of 11 of the 16 new apple varieties.

The birthplace of 'Huvitus' has been documented in detail in local municipal history. Locally-accepted oral tradition recognizes the person who sowed the seeds, sowing time and acquisition place of the seeds (apple fruit). As is typical of that time, a tenant farmer collected apple fruits from an old derelict orchard nearby which was the largest orchard in Finland in the 1800s and planted those seeds to his small garden in Yläne municipality (60° 88'N, 22° 34'E). The year of seed sowing is remembered, surprisingly, as exactly 1895. The tenant farmer was weak-eyed and did not write down notes about his small garden. The oral tradition of the birth and of this exceptionally early maturing tree with tasty fruits remained strong because of tight local social networks. The needy tenant was good at socializing with residents of the alcohol rehabilitation clinic, which was established on the manor estate, the very same manor estate whose derelict orchard he had collected apple fruits from. Clinic residents were upper-class educated people from towns. Interestingly, locals disputed who was the first to recognize the value of the tree, make the first crafting and recommend it for cultivation.

Locals knew 'Huvitus' very well since the variety grows in many home-gardens. In addition, they were familiar with where the original site was situated. However, it was somewhat surprising for many locals that the mother tree is still alive. It grows in an abandoned former garden; the farmhouse was pulled doen in 1960, and forest trees took over the garden and were later cut down. The research group with local informants found a small stub stem with one branch in the abandoned garden. The size of the tree did not show an age of over 100 years. However, its genotype identifier was the same as 'Huvitus' in nurseries and in the apple collection utilized in the breeding program (Figure 1). The sample for the rootstock genotype identifier was analyzed from a piece of root below the ground. The rootstock had the very same genotype identifier as the other 'Huvitus' named samples (Figure 1). This proved that the tree is a trueness-to-type mother tree. Later, in interviews with locals, it turned out that it has had several branches (ramifying).

Afterwards, the local heritage association established a local conservation in situ in-garden site for it. It has a fence and a named nursing person. It regenerates itself by growing a new shoot from the root, which will one day be the main stem.

*3.4. Rearranging the National Germplasm Apple Collection*

The multi-approach trueness-to-type process and its results were not only utilized in renewing the national germplasm apple collection. In addition, the criteria with which national local apple varieties were accepted into the national collection was built on knowledge achieved in this process. The renewed criteria were accepted in discussions in the fruit expert group of the Finnish national genetic resources program where varieties and their accessions were also chosen to be added to the clonal collection. It was agreed that preserved national local apple varieties need to meet (a) a criterion of genetic and morphological difference and a minimum of two criteria of following (b) at least a 50-year-long documented (i.e., written and/or believable memory knowledge) cultivation history showing adaptation to local growing conditions (e.g., winter and/or frost hardiness); (c) a seed born mother tree has, or had been in the past, spread to at least local cultivation indicating some use value (e.g., utilized as fresh, in cooking; storage durability) and recognition among home gardeners, professional growers, nurseries; (d) it originates in Northern Finland (> IV growing zone) and meets the criteria a-c; (e) it has a special cultural-historic value (e.g., a special birth place, persons linked to it); (f) there exists an already known special use value for breeding, product development, cultivation (e.g., organic).

In the mid-2010s, some accessions in the national germplasm apple collection were infected by apple proliferation fytoplasma (*Candidatus Phytoplasma mali*). The in situ in-garden inventory results provided acquisition sources of pure plant material from home-gardens, commercial orchards and nurseries. During the years 2017–2020 the main Finnish germplasm apple collection is being re-established to a new site in Jokioinen research station (60° 48′N, 02° 29′E) of the Natural Resources Institute Finland, 80 kilometers to the northeast of the earlier main collection site. The total number of preserved apple varieties is 166 (year 2018) of which 97 are Finnish local varieties, 17 are Finnish bred new varieties, and 40 are foreign varieties having long cultivation history in Finland (of which 14 are Swedish, 16 are Russian, two are Danish, three are Estonian, and five are from the USA). Compared to the accessions in the years 2002–2005, the number that are of Finnish origin have doubled and the collection size has been reduced by one half. The collection has several unknown old accessions the origins of which we were unable to confirm. Moreover, we were unable to locate about 10 Finnish local apples that we wanted to search for in situ in-garden. They remain for future repeated in situ in-garden inventories to achieve trueness-to-type varieties and their accessions.

## 4. Discussion

Original Finnish apple germplasm is born at the final frontier of apple cultivation thus containing unique adaptation, created in those local varieties that have been able to survive harsh conditions. These local varieties may contain unique or rare gene variants or set of genes that might be important for future, yet unknown, needs. Maximization of use (use being also a way of active conservation) of this unique germplasm requires well defined and thorough characterization. Historically and traditionally, characterization of germplasm has been, and still is performed including different levels: descriptions of taxonomy, biogeography, morphology and agronomic traits [24,35] while during recent few decades also biochemical and genetic evaluation are done [35]. A well characterized collection is a pre-requisite for rationalization, efficacy in management and sustainable use including selection and breeding of apple. During our work on filling the gaps and verifying the identities of Finnish local varieties through the presented multi-approach trueness-to-type process, we found out that ultimate condition for being able to perform good characterization at different levels is, actually, to have correctly identified varieties and evidence-based reasoning why a certain accession is finally included into the national apple germplasm collection.

As already mentioned, the extensive European-wide or even world-wide apple database would significantly aid identification process especially having in mind genetic marker features (revealing samples with identical fingerprints by comparative analyses). In order to effectively use markers, Baric et al. [38,39] proposed to: establish a genetic database that will contain marker fingerprints of

well described reference cultivars from different collections; and to use at least three accessions from independent collections to confirm cultivar i.e., for reliable determination of unidentified samples. However, this showed to be challenging in case of old local varieties. We noticed that, in addition to the traditional, molecular and genetic evaluation of apple collections, an extensive and laborious group of experts' pre-work of historical data and socio-economic conditions research is missing. As shown by this study, this is enabling to trace apple identities, especially old and neglected ones, back to its roots i.e., back to its birth. Without historical evidence that will be verified through iteration of different levels of characterization and evaluation, and in case where pedigree is not known or not verified we can claim only duplicates but no trueness-to-type. Having pan-apple microsatellite database would, without any doubt, be of utmost importance to have in order to serve genetic diversity analyses, identification of duplicates and revealing identities but special attention has to be paid to the local varieties (landraces) as they will serve for filling the gaps and strengthen trueness-to-type analyses. Finally, having trueness-to-type proved old local apple varieties should be important goal nationally, and this approach modified according local needs might be useful tool in such a search.

The multi-approach trueness-to-type analysis of the named local varieties revealed several missing accessions and accessions that were incorrectly named (false accessions) in the national clonal collection. Furthermore, several accessions were verified as trueness-to-types (i.e., compared to at least one sample of the same name with differing origin). Coherent background data of a variety's historic origin was provided. The results of our analysis have contributed to the use of The Finnish National Genetic Resources Program, which has enabled us to rationalize clonal collection management, and to renew the criteria for selecting national local varieties in terms of preservation and improving its germplasm value.

The multi-approach trueness-to-type data, including historic origin data, was directly utilized to promote the use, accessibility and traceability of local apple varieties by using it in writing the officially-recognized descriptions of varieties for horticultural planting material in Finland during years 2017–2019 [14] following the EU legislation on the marketing of fruit plant propagating material and fruit plants intended for fruit production (Council Directive 2008/90/EC) and registration of suppliers and of varieties in the common list of varieties (Commission Implementing Directive 2014/97/EU.) Two thirds of the local apple varieties (68 varieties) that were accepted to the national clonal collection for long-term preservation have been registered [14] (the Finnish database will be merged with the NordGen GRIN database in 2020). They are marketed by specialized nurseries mainly for home gardeners and to some extent for commercial apple production, especially outside the southernmost area of Finland.

The multi-approach trueness-to-type analysis also resulted in an extensive list of geographical locations (over 100 different locations) of the studied old apple trees in situ in-garden (no named seed born, Finnish local apple varieties, foreign originating varieties) from the most southern parts of Finland (60° 29′N, 21° 56′E) up to the most northern areas (64° 88′N, 28° 47′E) where apples can be grown. In addition to the fact they are in use in various kinds of gardens, these can serve as a known backup reserve for the national clonal collection managed by various actors such as private home-gardens, specialized nurseries, public gardens, apple variety collectors, and professional and semi-professional apple farmers.

The data of trueness-to-type varieties can facilitate characterization studies such as nutrition, resistance to abiotic and biotic stresses for the use of future breeding tasks, and nutrition and other quality properties for niche products. The data of historic origin and sites can be utilized as branding [40].

The data produced with this multi-approach method can increase the interest of new stakeholders to use old apple varieties, such as museums renovating or establishing orchards [10,41] as well as public and private historic garden owners [37], local heritage associations, and rural and home economics associations [11]. This strengthens genetic diversity locally, and promotes general awareness and multiple ways of use.

In summary, during the years 2002–2005 [4], the Finnish national germplasm apple collection contained accessions of 38 Finnish local varieties. After the iterative evaluation process described in this article was implemented on a set of targeted local apple varieties, a collection was extended for additional accessions of 59 Finnish local varieties. This currently contains a total number of 97 varieties.

**Author Contributions:** Conceptualization, M.H., L.B.; writing M.H., L.B.; inventory, M.H.; data analysis inventory M.H.; data analysis SSR, L.B.; funding acquisition, M.H.

**Funding:** This research had several financiers. Data collation were co-funded by: the PGR Secure–Novel characterization of crop wild relative and landrace resources and a basis for improved crop breeding" under the EU Seventh Framework Programme [GA no. 266394], Finnish Cultural Foundation, Uusimaa Regional Fund, MTT Agrifood Research Finland, Finnish National Plant Genetic Resources Programme. SSR analysis was funded by Finnish National Plant Genetic Resources Programme. Data analysis completed in the project Farmer's Pride - Networking, partnerships and tools to enhance in situ conservation of European plant genetic resources" under the Horizon 2020 Framework Programme, [GA no. 774271].

**Acknowledgments:** We thank Kristiina Antonius and Pirjo Tanhuanpää for providing dna analysis data from earlier years. Particular gratitude goes to Hilma Kinnanen for providing her wide expertise on apple variety phenotyping, unique historic data and extensive apple grower network. Sirpa Moisander offered excellent technical assistance with SSR. The very special compliments go to the numerous private persons growing old apple trees and local apple varieties who provided memory knowledge and plant samples in Finland.

**Conflicts of Interest:** The authors declare no conflict of interests.

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
