# Peer review of "How to Discover Traditional Varieties and Shape in a National Germplasm Collection: The Case of Finnish Seed Born Apples (Malus × domestica Borkh.)"

_sustainability, doi:10.3390/su11247000_

Round 1

Reviewer 1 Report

I read with interest the manuscript entitled: “How to discover traditional varieties and shape national germplasm collection: A case of Finnish seed born apples (Malus x domestica Borkh.)”. The paper surely has merits, nevertheless I feel it should not be accepted in its current form and needs a lot of work to be ready for publication. The topic is extremely interesting: integrating different approaches to recognize and better conserve local varieties of crops. The English language is certainly below par and the manuscript does need the editing of a native speaker. There are several spelling mistakes as well as a bad use of punctuation.  The important topic is not well presented in the introduction as well not very well discussed in all its possible implications in the discussion. For those reason I think that the authors should deeply revise the paper before resubmission.

General comment: The paper needs an urgent revision from a native speaker. The logical structure of the sentences is very often not clear and there are several grammatical and spelling mistakes. There is an extreme lack of punctuation.

Abstract: Several things are not clear while reading the abstract, what are the aims of this research? was there any research hypothesis? what are the main findings?

Introduction: The poor English and lack of punctuation made this paragraph very difficult to follow and understand. Overall, I feel that the introduction does not give a satisfactory overview of the problem to the reader. The integration of genetic and historical data is an important issue in planning the conservation of PGR at national level. I think that a more in-depth explanation of the problem (i.e. importance of the integration of the different methodologies and weaknesses of using just one of the approaches) is needed also taking in consideration other researches beyond the Finnish experience. I feel that an introduction about the importance of conserving plant genetic resources and especially in situ conservation is missing.

Material and Methods: Once again, the linguistic issue makes very difficult to follow this section. Table 2 is unreadable.

Results: Same here, many spelling mistakes, repetitions and poor English made the all section difficult to follow. Figure 1 is unreadable. The five types illustrated at lines: 214-216 should be accompanied by a definition. How many example of each type did you encounter?   It would be really interesting and beneficial of the reader if the authors could provide some images of the varieties discussed and compared in the results. The case-study provided are very interesting but I fell that the whole picture gets lost sometimes,  a more quantitative view of the result is missing or not well underlined.

Discussion: The topic of the paper is really interesting (integrating genetic, historical and social aspects in landraces recognition and conservation), I feel that the results (and the nice case-studied presented), should be discussed also taking in consideration the impact of the proposed method in a more global way (beyond the Finnish experience). Produce a list of landraces and cultivar of interest for conservation is a priority for all European countries and several approaches and methods could be followed to achieve the goal of having better conserved PGR at continental level.

Author Response

Response to Reviewer 1 Comments

I read with interest the manuscript entitled: “How to discover traditional varieties and shape national germplasm collection: A case of Finnish seed born apples (Malus x domestica Borkh.)”

Point 1: General comment: The paper needs an urgent revision from a native speaker. The logical structure of the sentences is very often not clear and there are several grammatical and spelling mistakes. There is an extreme lack of punctuation.

Response 1: The revised manuscript has gone through an English editing, the service provided by MDPI. 

Point 2: Abstract: Several things are not clear while reading the abstract, what are the aims of this research? was there any research hypothesis? what are the main findings?

Response 2: The abstract is rewritten. Aims, hypothesis and main findings are described in the abstract; as well as properly mentioned in the rest of the manuscript e.g. at the end of the Introduction. 

Reviewer 1 comments on Introduction

Introduction: The poor English and lack of punctuation made this paragraph very difficult to follow and understand. Overall, I feel that the introduction does not give a satisfactory overview of the problem to the reader. The integration of genetic and historical data is an important issue in planning the conservation of PGR at national level. I think that a more in-depth explanation of the problem (i.e. importance of the integration of the different methodologies and weaknesses of using just one of the approaches) is needed also taking in consideration other researches beyond the Finnish experience. I feel that an introduction about the importance of conserving plant genetic resources and especially in situ conservation is missing.

Point 3: Introduction: The poor English and lack of punctuation made this paragraph very difficult to follow and understand. 

Response 3: The revised manuscript has gone through an English editing, the service provided by MDPI. 

Point 4: Introduction: Overall, I feel that the introduction does not give a satisfactory overview of the problem to the reader.

Response 4: Introduction is supplemented with relevant references,  the focus of the study described more precisely.  The observed need to gain well described, proof-rich samples for the trueness-to-type analysis of old heirloom apple varieties is showed in the introduction.

Point 5: Introduction: I think that a more in-depth explanation of the problem (i.e. importance of the integration of the different methodologies and weaknesses of using just one of the approaches) is needed also taking in consideration other researches beyond the Finnish experience. 

Response 5: Introduction is expanded by  in-depth explanation of the problem and provided with extensive number of reference researches carried out especially in Europe regarding genetic evaluations of apple germplasm collections.  

Point 6: Introduction:  I feel that an introduction about the importance of conserving plant genetic resources and especially in situ conservation is missing.

Response 6: Introduction is expanded by pointing out the importance to gain the trueness-to-type heirloom varieties to the apple germplasm collection. The acquisition material (old trees of the named variety) can be still  situated in situ on-garden (i.e. historic orchards). In discussion we show how the received  in situ on-garden inventory data can be used also in planning in situ conservation activities. The use (and their availability in present-day nurseries)  of heirloom varieties is actually the most desirable way of organizing in situ conservation (esp. of clonally propagated plants)   

Point 7: Material and Methods: Once again, the linguistic issue makes very difficult to follow this section. 

Response 7: The revised manuscript has gone through an English editing, the service provided by MDPI. 

Point 8: Material and Methods: Table 2 is unreadable.

Response 8:  Table 2 Data groups of in situ on-garden inventory  is off and its content incorporated in the Material and Methods and Results body text as well as the new Table 1 Names, origin, acquisition, historical and characterization data of 12 apple varieties classified into 5 cases (…) 

Reviewer 1 comments on Results

Same here, many spelling mistakes, repetitions and poor English made the all section difficult to follow. Figure 1 is unreadable. The five types illustrated at lines: 214-216 should be accompanied by a definition. How many example of each type did you encounter?   It would be really interesting and beneficial of the reader if the authors could provide some images of the varieties discussed and compared in the results. The case-study provided are very interesting but I fell that the whole picture gets lost sometimes,  a more quantitative view of the result is missing or not well underlined.

Point 9: Results: Same here, many spelling mistakes, repetitions and poor English made the all section difficult to follow.  

Response 9: The revised manuscript has gone through an English editing, the service provided by MDPI. 

Point 10: Results: Figure 1 is unreadable. 

Response 11: Figure 1 is  improved with layout quality. 

Point 10: Results:The five types illustrated at lines: 214-216 should be accompanied by a definition. 

Response 11: We have added more precise titles of the five types  (eg. genome size to diploid-triploid pairs) which all demonstrate the trueness-to-type process by combining morphological, fingerprinting and historic origin identifies.

Point 11: Results: How many example of each type did you encounter? 

Response 11: The first type (genome size, now renamed as diploid-triploid pair) included 2 pairs in all data. Both of them are presented in the Result section. The second type (renamed foreign varieties) included two cases in all data which translated names did not reveal their origin or the old pomology literature did not reveal their origin. Both of them  are presented in the Result section. The third type (several candidates for trueness-to-type) were numerous, approx. every fifth sample was a wrongly named. In the list of local varieties desirable to include to tho the trueness-to-type analysis, one third were rare or very rare belonging to the category of the fourth type (verifying a very rare local variety). Concerning the fifth type (trueness-to-type mother tree), we have identified in total 10 mother trees in the in situ on-garden inventory. In one of these cases the size of the tree did not show the age of memorized age (in the presented case over 100 years) and to gain verification of the trueness-to-type mother tree the SSR analysis was made from its rootstock.

Point 12: Results: It would be really interesting and beneficial of the reader if the authors could provide some images of the varieties discussed and compared in the results.

Response 12: Photos of apples are provided in the new Table 1 Names, origin, acquisition, historical and characterization data of 12 apple varieties classified into 5 cases (…) 

Point 13: Results:The case-study provided are very interesting but I fell that the whole picture gets lost sometimes,  a more quantitative view of the result is missing or not well underlined.

Response 13: We have provided the new Table 1 Names, origin, acquisition, historical and characterization data of 12 apple varieties classified into 5 cases (…) which summarize used data and provide some quantitative data of samples analysed. We have done some distillations to the text. 

Point 14: Discussion: The topic of the paper is really interesting (integrating genetic, historical and social aspects in landraces recognition and conservation), I feel that the results (and the nice case-studied presented), should be discussed also taking in consideration the impact of the proposed method in a more global way (beyond the Finnish experience). Produce a list of landraces and cultivar of interest for conservation is a priority for all European countries and several approaches and methods could be followed to achieve the goal of having better conserved PGR at continental level.

Response 14: We have extended the discussion section and provided discussion of benefits of the presented multi-approach trueness-to-type process for gaining a well-characterized national apple germplasm collection which improves the value of accessions and facilitate their use among different stakeholders. 

Reviewer 2 Report

The presented manuscript is very beneficial in its focus to rationalize and improve the germplasm value of the Finnish national apple clonal collection for long term clonal preservartion. The authors used both historic traditional data and SSR methods to evaluate apple samples. In particular, the aim was to detect duplicates and identify the truth of the varieties. The studied material were from Natural Resources Institute Finland, Luke and from a special in situ on garden inventory. I think that the methodological procedure using microsatellite markers to identify the set of unique samples was correct and very important results were obtained. But part “Results”contain too many texts and detailed descriptions especially part “Renamed foreign varieties”, I would recommend shortening the text, possibly putting some information into the discussion and organizing the results in tables for greater clarity. I expected the results to be reported according to the methodology to two parts Evaluation of national apple collection and In situ on-garden inventory. In the abstract I would recommend adding a specific number of tested varieties.

In dendrogram I did not find triploid varieties Turso and Eppulainen, could you explain, please (according to line 223 – 224)?

Line: 263 Explanation of codes, see Table 1. I think that this is not correct table number.

Author Response

Response to Reviewer 2 Comments

The presented manuscript is very beneficial in its focus to rationalize and improve the germplasm value of the Finnish national apple clonal collection for long term clonal preservartion. The authors used both historic traditional data and SSR methods to evaluate apple samples. In particular, the aim was to detect duplicates and identify the truth of the varieties. The studied material were from Natural Resources Institute Finland, Luke and from a special in situ on garden inventory. I think that the methodological procedure using microsatellite markers to identify the set of unique samples was correct and very important results were obtained

Point 1: “Results”contain too many texts and detailed descriptions especially part “Renamed foreign varieties”, I would recommend shortening the text, possibly putting some information into the discussion and organizing the results in tables for greater clarity.

Response 1: We have provided the new Table 1 Names, origin, acquisition, historical and characterization data of 12 apple varieties classified into 5 cases (…) which summarize used data and provide some quantitative data of samples analysed. We have done some distillations to the text. 

Point 2: Results: I expected the results to be reported according to the methodology to two parts Evaluation of national apple collection and In situ on-garden inventory. 

Response 2: We have rearranged both the Methods and Materials section and the Result section to get more comprehensiveness to the text. The focus of the article is to demonstrate the evaluation of national as an iterative process of combining multi-approached methods and data. This process is highlighted in the case study of five types.

Point 3: In the abstract I would recommend adding a specific number of tested varieties.

Response 3: The abstract is rewritten. Quantitative data is added. 

Point 4: Results: In dendrogram I did not find triploid varieties Turso and Eppulainen, could you explain, please (according to line 223 – 224)?

Response 4:  The triploid varieties is added to Fig 1 Dendrogram 

Point 5: Results: Line: 263 Explanation of codes, see Table 1. I think that this is not correct table number.

Response 5: Table 4 Characteristics identifiers and historic origin data of Kavlås, Turso, Grenman and Eppulainen is off and the content added with explanation of codes to the new Table 1 Names, origin, acquisition, historical and characterization data of 12 apple varieties classified into 5 cases (…) 

Reviewer 3 Report

Manuscript “How to discover traditional varieties and shape national germplasm collection: A case of Finnish seed born apples (Malus x domestica Borkh.)”, is timely and required for the apple researcher community. However, I have several major concerns which can be addressed to make the article more appropriate to publish in Plants

Abstract:

The abstract contains more background information compared to the findings of the present study, it required improvement. I suggest to cutdown introductory portions to only two sentences. And then describe significant results and conclusive points.

Line 10-11: sentence “major global crop of economic importance globally and regionally” modified as “major global crop of economic importance worldwide and regionally”

Line 12-17: sentence not clear re-write to make it more meaningful

Introduction:

Line 33: the sentence “in Finland apple trees are grown on the frontier of their northern growing limits” its complete copy from abstract section (Line-11-12), such redundancy should be removed from the entire manuscript.

Line 35: In the abstract it is mentioned northern growing limit, here it is southwest?

Line 37-39: English improvements required

Line 52-53: Give your own finding reference, otherwise remove it

Materials method:

Line 78: give space between “(Kinnanen, Antonius 2006)” and “it was”

Line 83-87: It’s better to write the exact methodology used and data used with reference, it contains more background information, it seems like the introduction.

Line 89-91: Materials and method do not require such text, please remove it “Meurman had a professorship 89 in horticulture in Agricultural Research Centre in 1935-1959 and during the time 315 varieties were 90 under pomology observations to Piikkiö [13]” just mentioned materials which authors have used.

The overall style of methodology writing does not look scientific, please follow the journal guideline and materials method writing style, it seems authors are writing introduction again, additionally methodology is much longer.

Results:

Line 90-93: it should be part of the discussion

Line 99-101: What is this??

Fig. 1: If possible picture quality should be increased.

Results should be more concise and focused on finding only, in general result does not require references, if still, it is not necessary.

Discussion:

Compared to findings the discussion section is too much shorter, authors should describe their findings with a proper citation in discussion in more detail.

Author Response

Response to Reviewer 3 Comments

Manuscript “How to discover traditional varieties and shape national germplasm collection: A case of Finnish seed born apples (Malus x domestica Borkh.)”, is timely and required for the apple researcher community. However, I have several major concerns which can be addressed to make the article more appropriate to publish in Plants

Reviewer 3 comments on Abstract

Point 1: Abstract:  The abstract contains more background information compared to the findings of the present study, it required improvement. I suggest to cutdown introductory portions to only two sentences. And then describe significant results and conclusive points.

Response 1: The abstract is rewritten. Aims, hypothesis and main findings are described in the abstract. 

Point 2: Abstract: Line 10-11: sentence “major global crop of economic importance globally and regionally” modified as “major global crop of economic importance worldwide and regionally”

Response 2: The abstract is rewritten. The sentence is re-formulated: Cultivated apple (Malus × domestica Borkh.) is the major global crop of economic importance globally and regionally. 

Point 3: Abstract: Line 12-17: sentence not clear re-write to make it more meaningful

Response 3: The abstract is rewritten. The sentences are re-formulated: It is currently, and was also in the past, the main commercial fruit in the northern European countries. In Finland, apple trees are grown on the frontier of their northern growing limits. Because of these limits, growing an apple tree from a seed was discovered in practice to be the most appropriate method to get trees that bear fruit for people in the north. This created a unique culturally and genetically rich native germplasm to meet the various needs of apple growers and consumers from the late 1800s to the mid-1900s. 

Reviewer 3 comments on Introduction

Point 4: Introduction: Line 33: the sentence “in Finland apple trees are grown on the frontier of their northern growing limits” its complete copy from abstract section (Line-11-12), such redundancy should be removed from the entire manuscript.

Response 4: The abstract is rewritten. Redundancy lines are removed from the manuscript and replaced with another expression. 

Point 5: Introduction: In the abstract it is mentioned northern growing limit, here it is southwest?

Response 5: The abstract is rewritten. Main present-day commercial production of apples in Finland is limited to the southwest of the country. The same was in the past, first apple trees were documented to be grown in particular gardens of southern Finland. However, local apple varieties showing adaptation to more northern conditions can still be found especially in home-gardens.  

Point 6: Introduction: Line 37-39: English improvements required

Response 6: The revised manuscript has gone through an English editing service provided by MDPI. 

Point 7: Introduction: Line 52-53: Give your own finding reference, otherwise remove it

Response 7: The sentence (While analyzing Finnish apple accessions genotype identifiers gained with SSR markers we noticed the importance of samples in analysis) is removed.

Reviewer 3 comments on Materials and Methods

Point 8: Materials and methods: give space between “(Kinnanen, Antonius 2006)” and “it was”

Response 8: The revised manuscript has gone through an English editing service provided by MDPI.

Point 9: Materials and methods: Line 83-87: It’s better to write the exact methodology used and data used with reference, it contains more background information, it seems like the introduction.

Response 9: Material and methods section is revised. The lines (Various methods and data were utilized in evaluation. Molecular analysis (SSR markers) was used to reveal duplicates and to assist in identifying trueness-to-types. Morphological (esp. shape, ground and over color, aperture of lobules, flowering and ripening time) with applied UPOV standards were observed from fresh fruit samples; and the received observation data compared to various variety catalogues (esp. pomology books) to identify trueness-to-types and synonym names.) are revised to more exact description of the used methods and materials.

Point 10: Materials and methods: Line 89-91: Materials and method do not require such text, please remove it “Meurman had a professorship 89 in horticulture in Agricultural Research Centre in 1935-1959 and during the time 315 varieties were 90 under pomology observations to Piikkiö [13]” just mentioned materials which authors have used.

Response 10: The sentence is removed from Materials and Methods section. It is rephrased in introduction: The Finnish breeding program for cultivated apple was established in 1958 on the basis of extensive pomological observations (in total 315 apple varieties studied during 1935-1958) [] at the Agricultural Research Center (current Natural Resources Institute Finland) situated in Piikkiö in the southwest of Finland (60° 25′ N, 22° 31′E).  

Point 11: Materials and methods:The overall style of methodology writing does not look scientific, please follow the journal guideline and materials method writing style, it seems authors are writing introduction again, additionally methodology is much longer.

Response 11: Material and methods section is revised and text is compressed. 

Reviewer 3 comments on Results

Point 12: Results: Line 90-93: it should be part of the discussion

Response 12:  The line 90-93 was in the Material and methods section and it is unclear to which the reviewer refers to. However, Material and methods is revised and text is compressed. As well as major changes in the Results and Discussion sections. 

Point 13: Results: Line 99-101: What is this??

Response 13: The line 99-101 was in the Material and methods section and it is unclear which the reviewer refers to. However, Material and methods is revised and text is compressed. As well as major changes in the Result and Discussion sections. Moreover, the Table 1 Apple varieties in the Finnish pomology by Meurman is off and the content of the table is presented in the text of the Result section.

Point 14: Results: Fig. 1: If possible picture quality should be increased.

Response 14: The quality of Figure 1 dendrogram resolution is improved. 

Point 15: Results: Results should be more concise and focused on finding only, in general result does not require references, if still, it is not necessary.

Response 15: Results section is revised. Introduction text elements are removed and the references except one. In Diploid-triploid pairs: During the analyses of microsatellite fingerprints, we noticed that some accessions, e.g., Turso and Eppulainen, sometimes have three microsatellite alleles in the frame of one marker, which might be an indication that those accessions are triploids (Table 2.) This was already recognized when we analyzed a triploid variety, ‘Rambo’ [38]. 

Point 16: Discussion: Compared to findings the discussion section is too much shorter, authors should describe their findings with a proper citation in discussion in more detail.

Response 16: We have extended the discussion section with core references and provided discussion of benefits of the presented multi-approached trueness-to-type process for gaining a well-characterized national apple germplasm collection which improves the value of accessions and facilitate their use among different stakeholders.

Round 2

Reviewer 1 Report

I had the pleasure to review the revision of the paper: "How to Discover Traditional Varieties and Shape National Germplasm Collection: A Case of Finnish Seed Born Apples (Malus ×domestica Borkh.)" by Heinonen and Bitz. I feel that the manuscript really benefited from the extensive English linguistic revision. The authors made the effort to better contextualize their research and findings in the Introduction and Discussion sections. I will suggest to accept the paper.   

Reviewer 2 Report

I agree with corrected version of manuscript.

Reviewer 3 Report

The revised version of the MS looks appropriate.